# Existence of Mild Solutions for Hilfer Fractional Neutral Integro-Differential Inclusions via Almost Sectorial Operators

**Chandra Bose Sindhu Varun Bose** and **Ramalingam Udhayakumar** *

Department of Mathematics, School of Advanced Sciences, Vellore Institute of Technology, Vellore 632 014, Tamil Nadu, India
* Correspondence: udhayaram.v@gmail.com or udhayakumar.r@vit.ac.in

**Abstract:** This manuscript focuses on the existence of a mild solution Hilfer fractional neutral integro-differential inclusion with almost sectorial operator. By applying the facts related to fractional calculus, semigroup, and Martelli's fixed point theorem, we prove the primary results. In addition, the application is provided to demonstrate how the major results might be applied.

**Keywords:** Hilfer fractional system; neutral system; multi-valued maps; sectorial operators

**MSC:** 26A33; 34A08; 34K30; 47D09

## 1. Introduction

In modern mathematics, the fundamentals surrounding fractional computation and the fractional differential equation have taken center stage. The idea of fractional computation has now been put to the test in a wide variety of social, physical, signal, image processing, biological, control theory, engineering, etc., challenges. However, it has been demonstrated that fractional differential equations may be a valuable tool for describing a variety of situations. For many different types of realistic applications, fractional-order models are superior to integer-order models. The research articles [1–15] are concerned with the theory of fractional differential systems, and readers will find a number of fascinating findings about fractional dynamical systems. Please refer to [16–21] for more information.

Other fractional derivatives introduced by Hilfer [22] include the R-L derivative and Caputo fractional derivative. Many scholars have recently shown tremendous interest in this area, e.g., [23–25]; researchers have established their results with the help of Schauder's fixed point theorem. In [26–28], the authors worked on the existence and controllability of differential inclusions via the fixed point theorem approach. In references [29–31], the authors discussed the existence of a mild solution by using Martelli's fixed point theorem. As a result of these findings, we expand on the literature's earlier findings to a class of Hilfer fractional differential ($HFD$) systems in which the closed operator is almost sectorial.

In [32], M. Zhou, C. Li, and Y. Zhou studied the existence of mild solutions to Hilfer fractional differential equations with the order $\lambda \in (0,1)$ and type $\nu \in [0,1]$ in the abstract sense, as follows:

$$
\begin{aligned}
{}^{H}D_{0^+}^{\lambda,\nu}y(t) &= Ay(t) + g(t,y(t)),\ t \in (0,T], \\
I_{0^+}^{(1-\lambda)(1-\nu)}y(0) &= y_0,
\end{aligned}
$$

here, $A$ denotes the almost sectorial operator of the semigroup and the Schauder fixed point theorem is used.

In [33], Zhang and Zhou demonstrated the existence of fractional Cauchy problems using almost sectorial operators of the type,

$$^{L}D_{0+}^{q}x(t) = Ax(t) + f(t, x(t)) \ t \in [0, a],$$
$$I_{0+}^{(1-q)}x(0) = x_0,$$

where $^{L}D_{0+}^{q}$ is the $R-L$ derivative of order $q$, $0 < q < 1$, $I_{0+}^{(1-q)}$ is the $R-L$ integral of order $1-q$, $A$ is an almost sectorial operator on a complex Banach space. We refer the reader to [34–37] for information. These discoveries led us to extend past findings in the literature to Hilfer fractional Volterra–Fredholm integro-differential inclusions.

We will examine the following subject in the article: The almost sectorial operators are contained in the *HF* neutral integro-differential inclusion,

$$D_{0+}^{\kappa,\varepsilon}\left[y(\mathfrak{z}) - \mathcal{N}(\mathfrak{z}, y(\mathfrak{z}))\right] \in Ay(\mathfrak{z}) + \mathcal{G}\left(\mathfrak{z}, y(\mathfrak{z}), \int_0^{\mathfrak{z}} e(\mathfrak{z}, s, y(s)) ds\right), \qquad \mathfrak{z} \in \mathcal{J}' = (0, d], \quad (1)$$

$$I_{0+}^{(1-\kappa)(1-\varepsilon)}y(0) = y_0, \tag{2}$$

where $D_{0+}^{\kappa,\varepsilon}$ notates the *HFD* of order $\kappa$, $0 < \kappa < 1$, type $\varepsilon$, $0 \le \varepsilon \le 1$; and A is an almost sectorial operator of the analytic semigroup $\{T(\mathfrak{z}), \mathfrak{z} \ge 0\}$ on Y. State $y(\cdot)$ takes the value in a Banach space Y with norm $\| \cdot \|$. Let $\mathcal{J} = [0, d]$, $\mathcal{N} : \mathcal{J} \times Y$ be the appropriate function, $\mathcal{G} : \mathcal{J} \times Y \times Y \to 2^Y \setminus \{\emptyset\}$ be a non-empty, bounded, closed convex multi-valued map, $\mathcal{N} : \mathcal{J} \times Y \to Y$ and $e : \mathcal{J} \times \mathcal{J} \times Y \to Y$ are the appropriate functions.

This article is structured as follows: In Section 2, we present the fundamentals of fractional differential systems, semigroup, and closed linear operators. In Section 3, we present the existence of the required solution. In Section 4, we provide an application to demonstrate our main arguments and some inferences are established in the end.

## 2. Preliminaries

Here, we introduce some basic definitions, theorems, and lemmas that are applied to every part of the paper.

Let $\complement$ be the collection of all continuous functions from $\mathcal{J}$ to Y, where $\mathcal{J} = [0, d]$ and $\mathcal{J}' = (0, d]$ with $d > 0$. Take $\mathcal{X} = \{y \in \complement : \lim_{\mathfrak{z} \to 0} \mathfrak{z}^{1-\varepsilon+\kappa\varepsilon-\kappa\xi}y(\mathfrak{z})$ exists and finite $\}$, which is the Banach space and its norm on $\| \cdot \|_{\mathcal{X}}$, defined as $\|y\|_{\mathcal{X}} = \sup_{\mathfrak{z} \in \mathcal{J}'}\{\mathfrak{z}^{1-\varepsilon+\kappa\varepsilon-\kappa\xi}\|y(\mathfrak{z})\|\}$. Let $y(\mathfrak{z}) = \mathfrak{z}^{-1+\varepsilon-\kappa\varepsilon+\kappa\xi}u(\mathfrak{z})$, $\mathfrak{z} \in (0, d]$ then, $y \in \mathcal{X}$ *iff* $y \in \complement$ and $\|y\|_{\mathcal{X}} = \|y\|$. Moreover, define $B_P(\mathcal{J}) = \{y \in \complement \text{ such that } \|y\| \le P\}$.

**Definition 1** ([19])**.** *The left side of the R-L fractional integral of order $\kappa$ with the lower limit d for function $\mathcal{G} : [d, \infty) \to \mathbb{R}$ is presented by*

$$I_{d+}^{\kappa}\mathcal{G}(\mathfrak{z}) = \frac{1}{\Gamma(\kappa)} \int_d^{\mathfrak{z}} \frac{\mathcal{G}(w)}{(\mathfrak{z} - w)^{1-\kappa}} dw, \ \mathfrak{z} > 0, \ \kappa > 0,$$

*provided the right side is pointwise determined on $[d, +\infty)$, $\Gamma(\cdot)$ is the gamma function.*

**Definition 2** ([19])**.** *The left-sided R-L fractional derivative of order $\kappa > 0$, $m - 1 \le \kappa < m$, $m \in \mathbb{N}$, for a function $\mathcal{G} : [d, +\infty) \to \mathbb{R}$ is presented by*

$$^{L}D_{d+}^{\kappa}\mathcal{G}(\mathfrak{z}) = \frac{1}{\Gamma(m-\kappa)} \frac{d^m}{d\mathfrak{z}^m} \int_d^{\mathfrak{z}} \frac{\mathcal{G}(w)}{(\mathfrak{z} - w)^{\kappa+1-m}} dw, \ \mathfrak{z} > d,$$

*where $\Gamma(\cdot)$ is the gamma function.*

**Definition 3** ([19])**.** *The left-sided Caputo derivative of the type of order $\kappa > 0$, $m - 1 \leq \kappa < m$, $m \in \mathbb{N}$ for a function $\mathcal{G} : [d, +\infty) \to \mathbb{R}$, is defined as*

$$^{C}D_{d+}^{\kappa}\mathcal{G}(\mathfrak{z}) = \frac{1}{\Gamma(m - \kappa)} \int_{d}^{\mathfrak{z}} \frac{\mathcal{G}^{m}(w)}{(\mathfrak{z} - w)^{\kappa + 1 - m}}dw = I_{d+}^{m-\kappa}\mathcal{G}^{m}(\mathfrak{z}), \ \mathfrak{z} > d,$$

*where $\Gamma(\cdot)$ is the gamma function.*

**Definition 4** ([22])**.** *The left-sided HFD of order $0 < \kappa < 1$ and type $\varepsilon \in [0, 1]$, of function $\mathcal{G} : [d, +\infty) \to \mathbb{R}$, is defined as*

$$D_{d+}^{\kappa,\varepsilon}\mathcal{G}(\mathfrak{z}) = [I_{d+}^{(1-\kappa)\varepsilon}D(I_{d+}^{(1-\kappa)(1-\varepsilon)}\mathcal{G})](\mathfrak{z}).$$

**Remark 1** ([22])**.** *1.     If $\varepsilon = 0$, $0 < \kappa < 1$, and $d = 0$, then the HFD corresponds to the classical R-L fractional derivative:*

$$D_{0+}^{\kappa,0}\mathcal{G}(\mathfrak{z}) = \frac{d}{d\mathfrak{z}}I_{0+}^{1-\kappa}\mathcal{G}(\mathfrak{z}) = \ ^{L}D_{0+}^{\kappa}\mathcal{G}(\mathfrak{z}).$$

2.     *If $\varepsilon = 1$, $0 < \kappa < 1$, and $d = 0$, then the HFD corresponds to the classical Caputo fractional derivative:*

$$D_{0+}^{\kappa,1}\mathcal{G}(\mathfrak{z}) = I_{0+}^{1-\kappa}\frac{d}{d\mathfrak{z}}\mathcal{G}(\mathfrak{z}) = \ ^{C}D_{0+}^{\kappa}\mathcal{G}(\mathfrak{z}).$$

**Definition 5** ([38])**.** *For $0 < \xi < 1$,    $0 < \omega < \frac{\pi}{2}$, $\Theta_{\omega}^{-\xi}$ is the family of closed linear operators, the sector $S_{\omega} = \{v \in \mathbb{C} \backslash \{0\} \text{ with } |\arg v| \leq \omega\}$, and $\mathtt{A} : D(\mathtt{A}) \subset Y \to Y$, which satisfy*

(i)     $\sigma(\mathtt{A}) \subseteq S_{\omega}$ ;
(ii)     *For any $\omega < \delta < \pi \ \exists \ \Lambda_{\delta}$ is a constant, such that,*

$$\left\|(vI - \mathtt{A})^{-1}\right\| \leq \Lambda_{\delta}|v|^{-\xi}$$

*then $\mathtt{A} \in \Theta_{\omega}^{-\xi}$ is called an almost sectorial operator on $Y$.*

**Lemma 1** ([38])**.** *Let $0 < \xi < 1$ and $0 < \omega < \frac{\pi}{2}$, $\mathtt{A} \in \Theta_{\omega}^{-\xi}(Y)$. Then*

1.     $T(\mathfrak{z}_{1} + \mathfrak{z}_{2}) = T(\mathfrak{z}_{1}) + T(\mathfrak{z}_{2})$, *for any $\mathfrak{z}_{1}, \mathfrak{z}_{2} \in S_{\frac{\pi}{2}-\omega}^{0}$;*
2.     $\exists \ \Lambda_{0} > 0$ *is the constant, such that $\|T(\mathfrak{z})\|_{\mathbb{C}} \leq \Lambda_{0}\mathfrak{z}^{\xi-1}$, for any $\mathfrak{z} > 0$;*
3.     *The range $R(T(\mathfrak{z}))$ of $T(\mathfrak{z})$, $\mathfrak{z} \in S_{\frac{\pi}{2}-\omega}^{0}$ is contained in $D(\mathtt{A}^{\infty})$. Particularly, $R(T(\mathfrak{z})) \subset D(\mathtt{A}^{\theta})$ for all $\theta \in \mathbb{C}$ with $Re(\theta) > 0$,*

$$\mathtt{A}^{\theta}T(\mathfrak{z})\mathtt{y} = \frac{1}{2\pi i}\int_{\Gamma_{\gamma}} z^{\theta}e^{-\mathfrak{z}z}R(z; \mathtt{A})\mathtt{y}dz, \ for \ all \ \mathtt{y} \in Y,$$

*and, hence, $\exists$ is a constant $\Lambda' = \Lambda'(\beta, \theta) > 0$, such that*

$$\left\|\mathtt{A}^{\theta}T(\mathfrak{z})\right\|_{B(Y)} \leq \Lambda'\mathfrak{z}^{-\beta-Re(\theta)-1}, \ for \ all \ \mathfrak{z} > 0;$$

4.     *If $\theta > 1 - \xi$, then $D(\mathtt{A}^{\theta}) \subset \Sigma_{T} = \{y \in Y : \lim_{\mathfrak{z} \to 0+} T(\mathfrak{z})y = y\}$;*
5.     $R(\kappa', \mathtt{A}) = \int_{0}^{\infty} e^{-\kappa'\mathfrak{z}}T(\mathfrak{z})d\mathfrak{z}, \ \forall \ \kappa' \in \mathbb{C}$ *with $Re(\kappa') > 0$.*

*Consider the operator families* $\left\{ \mathcal{S}_\kappa(\mathfrak{z}) \right\}_{\mathfrak{z} \in S_{\frac{\pi}{2} - \omega}}$, $\left\{ \mathcal{Q}_\kappa(\mathfrak{z}) \right\}_{\mathfrak{z} \in S_{\frac{\pi}{2} - \omega}}$ *is defined as follows:*

$$\mathcal{S}_\kappa(\mathfrak{z}) = \int_0^\infty W_\kappa(\nu) T(\mathfrak{z}^\kappa \nu) d\nu,$$

$$\mathcal{Q}_\kappa(\mathfrak{z}) = \int_0^\infty \kappa \nu W_\kappa(\nu) T(\mathfrak{z}^\kappa \nu) d\nu,$$

*where* $W_\kappa(\beta)$ *is the Wright-type function:*

$$W_\kappa(\beta) = \sum_{n \in \mathbb{N}} \frac{(-\beta)^{n-1}}{\Gamma(1 - \kappa n)(n-1)!}, \qquad \beta \in \mathbb{C}. \tag{3}$$

*Let* $-1 < \iota < \infty$, $p > 0$, *the succeeding properties are satisfied.*

(a)　$W_\kappa(\theta) \geq 0$, 　$\mathfrak{z} > 0$;

(b)　$\int_0^\infty \theta^\iota W_\kappa(\theta) d\theta = \frac{\Gamma(1+\iota)}{\Gamma(1+\kappa\iota)}$;

(c)　$\int_0^\infty \frac{\kappa}{\theta^{(\kappa+1)}} e^{-p\theta} W_\kappa\left(\frac{1}{\theta^\kappa}\right) d\theta = e^{-p^\kappa}$.

**Theorem 1** ([19]). $\mathcal{S}_\kappa(\mathfrak{z})$ *and* $\mathcal{Q}_\kappa(\mathfrak{z})$ *are continuous in the uniform operator topology, for* $\mathfrak{z} > 0$, *for every* $c > 0$, *the continuity is uniform on* $[c, \infty)$.

**Definition 6** ([16]). *A multi-valued map* $\mathcal{G}$ *called u.s.c. on* $Y$ *if for each* $y_0 \in Y$ *the set* $\mathcal{G}(y_0)$ *is a non-empty, closed subset of* $Y$, *and if for each open set* $\mathcal{U}$ *of* $Y$ *containing* $\mathcal{G}(y_0)$, *there exists an open neighborhood* $\mathcal{V}$ *of* $y_0$, *such that* $\mathcal{G}(\mathcal{V}) \subseteq \mathcal{U}$.

**Definition 7** ([16]). $\mathcal{G}$ *is said to be completely continuous if* $\mathcal{G}(C)$ *is relatively compact for each bounded subset* $C$ *of* $Y$. *If a multi-valued map* $\mathcal{G}$ *is completely continuous with non-empty compact values, then* $\mathcal{G}$ *is upper semi-continuous if and only if* $\mathcal{G}$ *has a closed graph i.e.,* $y_m \to y_0, \mathfrak{z}_m \to \mathfrak{z}_0$, $\mathfrak{z}_m \in \mathcal{G}(y_m)$ *imply* $\mathfrak{z}_0 \in \mathcal{G}(y_0)$.

**Definition 8** ([16]). *A multi-valued mapping* $\mathcal{G} : Y \to 2^Y$ *is said to be condensing, if for any bounded subset* $D \subset Y$ *with* $\beta(D) \neq 0$, *we have* $\beta(F(D)) < \beta(D)$, *where* $\beta(\cdot)$ *denotes the Kuratowski measure of non-compactness, defined as follows:*

$$\beta(D) = \inf \left\{ d > 0 : D \text{ covered by a finite number of balls of radius } d \right\}.$$

**Lemma 2.** *System* (1)–(2) *is equivalent to an integral inclusion given by*

$$y(\mathfrak{z}) \in \frac{y_0 - \mathcal{N}(0, y(0))}{\Gamma(\varepsilon(1-\kappa) + \kappa)} \mathfrak{z}^{(1-\kappa)(\varepsilon-1)} + \mathcal{N}(\mathfrak{z}, y(\mathfrak{z})) + \frac{1}{\Gamma(\kappa)} \int_0^\mathfrak{z} (\mathfrak{z} - w)^{\kappa-1} \mathtt{A}\mathcal{N}(w, y(w)) dw$$

$$+ \frac{1}{\Gamma(\kappa)} \int_0^\mathfrak{z} (\mathfrak{z} - w)^{\kappa-1} \left[ \mathtt{A}y(w) + \mathcal{G}\left( w, y(w), \int_0^w e(w, s, y(s)) ds \right) dw \right].$$

**Definition 9.** *By a mild solution of the Cauchy problem* (1)–(2), *the function* $y(\mathfrak{z}) \in C(\mathcal{J}', Y)$ *satisfies*

$$y(\mathfrak{z}) = \mathcal{S}_{\kappa,\varepsilon}(\mathfrak{z}) \left[ y_0 - \mathcal{N}(0, y(0)) \right] + \mathcal{N}(\mathfrak{z}, y(\mathfrak{z})) + \int_0^\mathfrak{z} \mathcal{K}_\kappa(\mathfrak{z} - w) \mathtt{A}\mathcal{N}(w, y(w)) dw$$

$$+ \int_0^\mathfrak{z} \mathcal{K}_\kappa(\mathfrak{z} - w) \mathcal{G}\left( w, y(w), \int_0^w e(w, s, y(s)) ds \right) dw, \quad \mathfrak{z} \in \mathcal{J},$$

*where* $\mathcal{S}_{\kappa,\varepsilon}(\mathfrak{z}) = I_0^{\varepsilon(1-\kappa)} \mathcal{K}_\kappa(\mathfrak{z})$, $\mathcal{K}_\kappa(\mathfrak{z}) = \mathfrak{z}^{\kappa-1} \mathcal{Q}_\kappa(\mathfrak{z})$.

**Lemma 3** ([32]). *For any fixed $\nu > 0$, $\mathcal{Q}_\kappa(\nu)$, $\mathcal{K}_\kappa(\nu)$ and $\mathcal{S}_{\kappa,\varepsilon}(\nu)$ are linear operators, and for any $y \in Y$,*

$$\left\| \mathcal{Q}_\kappa(\mathfrak{z}) \right\| \leq L'\mathfrak{z}^{-\kappa+\kappa\xi}, \ \left\| \mathcal{K}_\kappa(\mathfrak{z})y \right\| \leq L'\mathfrak{z}^{-1+\kappa\xi}\|y\|, \ \left\| \mathcal{S}_{\kappa,\varepsilon}(\mathfrak{z})y \right\| \leq L''\mathfrak{z}^{-1+\varepsilon-\kappa\varepsilon+\kappa\xi}\|y\|,$$

*where*

$$L' = \Lambda_0 \frac{\Gamma(\xi)}{\Gamma(\kappa\xi)}, \ L'' = \Lambda_0 \frac{\Gamma(\xi)}{\Gamma(\varepsilon(1-\kappa) + \kappa\xi)}.$$

**Lemma 4** ([32]). *Let $\left\{ T(\mathfrak{z}) \right\}_{\mathfrak{z}>0}$ be equicontinuous, then $\left\{ \mathcal{Q}_\kappa(\mathfrak{z}) \right\}_{\mathfrak{z}>0}$, $\left\{ \mathcal{K}_\kappa(\mathfrak{z}) \right\}_{\mathfrak{z}>0}$, and $\left\{ \mathcal{S}_{\kappa,\varepsilon}(\mathfrak{z}) \right\}_{\mathfrak{z}>0}$ are strongly continuous, i.e., for any $y \in Y$ and $\mathfrak{z}_2 > \mathfrak{z}_1 > 0$,*

$$\left\| \mathcal{Q}_\kappa(\mathfrak{z}_2)y - \mathcal{Q}_\kappa(\mathfrak{z}_1)y \right\| \to 0, \ \left\| \mathcal{K}_\kappa(\mathfrak{z}_2)y - \mathcal{K}_\kappa(\mathfrak{z}_1)y \right\| \to 0$$
$$\left\| \mathcal{S}_{\kappa,\varepsilon}(\mathfrak{z}_2)y - \mathcal{S}_{\kappa,\varepsilon}(\mathfrak{z}_1)y \right\| \to 0, \ as \ \mathfrak{z}_2 \to \mathfrak{z}_1.$$

**Proposition 1** ([39]). *Let $\kappa \in (0,1), \mu \in (0,1]$ and for all $y \in D(\mathtt{A})$, there exists a $\Lambda_\mu > 0$, such that*

$$\left\| \mathtt{A}^\mu \mathcal{Q}_\kappa(\mathfrak{z})y \right\| \leq \frac{\kappa\Lambda_\mu\Gamma(2-\mu)}{\mathfrak{z}^{\kappa\mu}\Gamma(1+\kappa(1-\mu))}\|y\|, \ 0 < \mathfrak{z} < d.$$

**Lemma 5** ([40]). *Let $\mathcal{J}$ be a compact real interval and $\mathcal{P}_{bd,cv,cl}(Y)$ be the set of all non-empty, bounded, convex, and closed subsets of $Y$. Let $\mathcal{G}$ be the $L^1$-Carathéodory multi-valued map, measurable to $\mathfrak{z}$ for each $y \in Y$, u.s.c. to $y$ for each $\mathfrak{z} \in C(\mathcal{J}, Y)$, the set*

$$S_{\mathcal{G},y} = \left\{ g \in L^1(\mathcal{J}, Y) : g(\mathfrak{z}) \in \mathcal{G}\left( \mathfrak{z}, y(\mathfrak{z}), \int_0^w e(w, s, y(s))ds \right), \ \mathfrak{z} \in \mathcal{J} \right\}, \tag{4}$$

*is non-empty. Let $\mathrm{Y}$ be the linear continuous function from $L^1(\mathcal{J}, Y)$ to $\mathtt{C}$, then*

$$\mathrm{Y} \circ S_{\mathcal{G}} : \mathtt{C} \to \mathcal{P}_{bd,cv,cl}(\mathtt{C}), \quad y \to (\mathrm{Y} \circ S_{\mathcal{G}})(y) = \mathrm{Y}(S_{\mathcal{G},y}), \tag{5}$$

*is a closed graph operator in $\mathtt{C} \times \mathtt{C}$.*

**Lemma 6** (Martelli's fixed point theorem [17]). *Let $Y$ be a Banach space and $F : Y \to \mathcal{P}_{bd,cv,cl}(Y)$ be an upper semi-continuous and condensing map. If the set*

$$\mathcal{M} = \{y \in Y : \lambda y \in F(y) \text{ for some } \lambda > 1\}$$

*is bounded, then $F$ has a fixed point.*

## 3. Existence

We need the succeeding hypotheses:

$(H_1)$ The almost sectorial operator $\mathtt{A}$ produces an analytic semigroup $\mathtt{T}(\mathfrak{z})$, where $\mathfrak{z} \geq 0$ in $Y$ and $\|\mathtt{T}(\mathfrak{z})\| \leq M$, for some $M > 0$.

$(H_2)$ (a)　Let $\mathcal{G} : \mathcal{J} \times Y \times Y \to \mathcal{P}_{bd,cv,cl}(Y)$ be measurable to $\mathfrak{z}$ for each fixed $y \in Y$, upper semi-continuous to $y$ for each $\mathfrak{z} \in \mathcal{J}$, and each $y \in \mathtt{C}$, take

$$S_{\mathcal{G},y} = \left\{ g \in L^1(\mathcal{J}, Y) : g(\mathfrak{z}) \in \mathcal{G}\left( \mathfrak{z}, y(\mathfrak{z}), \int_0^w e(w, s, y(s))ds \right), \ \mathfrak{z} \in \mathcal{J} \right\},$$

　　　　　is non-empty.

　　　(b)　For $\mathfrak{z} \in \mathcal{J}$, $\mathcal{G}(\mathfrak{z}, \cdot, \cdot) : Y \times Y \to Y$, $e(\mathfrak{z}, s, \cdot) : Y \to Y$ are continuous functions and for each $y \in \mathtt{C}$, $\mathcal{G}(\cdot, y, \int e) : \mathcal{J} \to \mathcal{I}$ and $e(\cdot, \cdot, y) : \mathcal{I} \times \mathcal{J} \to Y$ are strongly measurable.

(c)　　There exists a function $\phi(\mathfrak{z}) \in C(\mathcal{J}', \mathbb{R}^+)$ satisfying

$$\lim_{\mathfrak{z} \to 0^+} \mathfrak{z}^{1-\varepsilon+\kappa\varepsilon-\kappa\xi} I_{0^+}^{\kappa\xi} \phi(\mathfrak{z}) = 0$$

$$\|\mathcal{G}(\mathfrak{z}, \mathfrak{z}_1, \mathfrak{z}_2)\| = \sup\left\{ \|\mathcal{G}\| : \mathcal{G}(\mathfrak{z}) \in \mathcal{G}\left(\mathfrak{z}, y(\mathfrak{z}), \int_0^{\mathfrak{z}} e(\mathfrak{z}, s, y(s)) ds\right) \right\}$$

$$\leq \phi(\mathfrak{z})\Phi(\|\mathfrak{z}_1\| + \|\mathfrak{z}_2\|).$$

for a.e. $\mathfrak{z} \in \mathcal{J}$ and $\mathfrak{z}_1, \mathfrak{z}_2 \in Y$, where $\Phi : \mathbb{R}^+ \to (0, \infty)$ is a continuous, additive, and non-decreasing function, satisfying $\Phi(\gamma_1(\mathfrak{z})(y)) \leq \gamma_1(\mathfrak{z})\Phi(y)$, where $\gamma \in C(\mathcal{J}', \mathbb{R}^+)$.

(d)　　There exists $\psi \in C(\mathcal{J}', \mathbb{R}^+)$, such that

$$\left\| \int_0^{\mathfrak{z}} e(\mathfrak{z}, s, y(s)) \right\| \leq \psi(\mathfrak{z}) \|y\| \text{ for each } \mathfrak{z} \in \mathcal{J}, \ y \in Y.$$

($H_3$) For any $\mathfrak{z} \in \mathcal{J}$, multi-valued map $\mathcal{N} : \mathcal{J} \times Y \to Y$ is a continuous function and there exists $\mu \in (0, 1)$, such that $\mathcal{N} \in D(\mathtt{A}^\mu)$ and all $y \in Y$, $\mathfrak{z} \in \mathcal{J}$, $\mathtt{A}^\mu \mathcal{N}(\mathfrak{z}, \cdot)$ satisfy the following:

$$\left\|\mathtt{A}^\mu \mathcal{N}(\mathfrak{z}, y(\mathfrak{z}))\right\| \leq M_g\left(1 + \mathfrak{z}^{1-\varepsilon+\kappa\varepsilon-\kappa\xi} \|y(\mathfrak{z})\|\right) \ and \ \left\|\mathtt{A}^{-\mu}\right\| \leq M_0, \ (\mathfrak{z}, y) \in \mathcal{J} \times Y.$$

($H_4$) $\mathcal{N}$ is completely continuous, and for any bounded set $D \subset \mathbb{C}$, the set $\{\mathfrak{z} \to \mathcal{N}(\mathfrak{z}, y(\mathfrak{z})), \ y \in D\}$ is equicontinuous in $Y$.

**Theorem 2.** *Assume that* $(H_1) - (H_4)$ *hold. Then the HF system* (1)–(2) *has a mild solution on* $\mathcal{J}$, *provided*

$$L' \int_0^{\mathfrak{z}} (\mathfrak{z} - w)^{\kappa\xi-1} \phi(\mathfrak{z})(1 + \psi(\mathfrak{z})) dw < \int_{M_1^*}^{\infty} \frac{du}{\Phi(u)},$$

*where*

$$M_1^* = d^{1-\varepsilon+\kappa\varepsilon-\kappa\xi}\left[ L'' d^{-1+\varepsilon-\kappa\varepsilon+\kappa\xi}(y_0 - M_0 M_g) + M_0 M_g(1 + P) \right]$$

*and* $y_0 \in D(\mathtt{A}^\theta)$ *with* $\theta > 1 - \xi$.

**Proof.** We define the multi-valued operator $\Psi : \mathcal{X} \to \mathcal{P}(\mathcal{X})$ by

$$\Psi(y(\mathfrak{z})) = \left\{ z \in \mathcal{X} : z(\mathfrak{z}) = \mathfrak{z}^{1-\varepsilon+\kappa\varepsilon-\kappa\xi}\left[ \mathcal{S}_{\kappa,\varepsilon}(\mathfrak{z})\left[y_0 - \mathcal{N}(0, y(0))\right] + \mathcal{N}(\mathfrak{z}, y(\mathfrak{z}))\right.\right.$$

$$+ \int_0^{\mathfrak{z}} (\mathfrak{z} - w)^{\kappa-1} \mathcal{Q}_\kappa(\mathfrak{z} - w) \mathtt{A} \mathcal{N}(w, y(w)) dw$$

$$\left.\left. + \int_0^{\mathfrak{z}} (\mathfrak{z} - w)^{\kappa-1} \mathcal{Q}_\kappa(\mathfrak{z} - w) \mathcal{G}\left(w, y(w), \int_0^w e(w, s, y(s)) ds\right) \right] dw, \ \mathfrak{z} \in (0, d] \right\}.$$

To show that the fixed point of $\Psi$ exists.
**Step:1** Convexity of $\Psi(y) \ \forall \ y \in B_P(\mathcal{J})$.
　　Let $z_1, z_2 \in \{\Psi y(\mathfrak{z})\}$ and $h_1, h_2 \in S_{\mathcal{G}, y}$ such that $\mathfrak{z} \in \mathcal{J}$. We know

$$z_i = \mathfrak{z}^{1-\varepsilon+\kappa\varepsilon-\kappa\xi}\left[ \mathcal{G}_{\kappa,\varepsilon}(\mathfrak{z})\left[y_0 - \mathcal{N}(0, y(0))\right] + \mathcal{N}(\mathfrak{z}, y(\mathfrak{z}))\right.$$

$$\left. + \int_0^{\mathfrak{z}} (\mathfrak{z} - w)^{\kappa-1} \mathcal{Q}_\kappa(\mathfrak{z} - w) \mathtt{A} \mathcal{N}(w, y(w)) dw + \int_0^{\mathfrak{z}} (\mathfrak{z} - w)^{\kappa-1} \mathcal{Q}_\kappa(\mathfrak{z} - w) h_i(w) dw \right], \quad i = 1, 2.$$

Let $0 \leq \lambda \leq 1$; then for each of $\mathfrak{z} \in \mathcal{J}$, we have

$$
\begin{aligned}
\lambda z_1 + (1-\lambda)z_2(\mathfrak{z}) = {}& \mathfrak{z}^{1-\varepsilon+\kappa\varepsilon-\kappa\xi} \bigg( \mathcal{S}_{\kappa,\varepsilon}(\mathfrak{z}) \big[ \mathrm{y}_0 - \mathcal{N}(0,\mathrm{y}(0)) \big] + \mathcal{N}(\mathfrak{z},\mathrm{y}(\mathfrak{z})) \\
& + \int_0^{\mathfrak{z}} (\mathfrak{z}-w)^{\kappa-1} \mathcal{Q}_\kappa(\mathfrak{z}-w) \mathtt{A}(w,\mathrm{y}(w)) dw \bigg) \\
& + \mathfrak{z}^{1-\varepsilon+\kappa\varepsilon-\kappa\xi} \int_0^{\mathfrak{z}} (\mathfrak{z}-w)^{\kappa-1} \mathcal{Q}_\kappa(\mathfrak{z}-w) \big[ \lambda h_1(w) + (1-\lambda)h_2(w) \big] dw.
\end{aligned}
$$

We know that $\mathcal{N}$ has a convex value, then $S_{\mathcal{G},\mathrm{y}}$ is convex. So, $\lambda h_1 + (1-\lambda)h_2 \in S_{\mathcal{G},\mathrm{y}}$. Therefore,

$$
\lambda z_1 + (1-\lambda)z_2 \in \Psi \mathrm{y}(\mathfrak{z}),
$$

hence $\Psi$ is convex.

**Step 2:** Boundness of $\Psi$ on $B_P(\mathcal{J})$. Consider, $\forall \, \mathrm{y} \in B_P(\mathcal{J})$, we have

$$
\begin{aligned}
\big\| z(\mathfrak{z}) \big\| \leq {}& \sup \mathfrak{z}^{1-\varepsilon+\kappa\varepsilon-\kappa\xi} \bigg\| \mathcal{S}_{\kappa,\varepsilon}(\mathfrak{z}) \big[ \mathrm{y}_0 - \mathcal{N}(0,\mathrm{y}(0)) \big] + \mathcal{N}(\mathfrak{z},\mathrm{y}(\mathfrak{z})) \\
& + \int_0^{\mathfrak{z}} (\mathfrak{z}-w)^{\kappa-1} \mathcal{Q}_\kappa(\mathfrak{z}-w) \mathtt{A}\mathcal{N}(w,\mathrm{y}(w)) dw \\
& + \int_0^{\mathfrak{z}} (\mathfrak{z}-w)^{\kappa-1} \mathcal{Q}_\kappa(\mathfrak{z}-w) \mathcal{G}\left( w,\mathrm{y}(w),\int_0^w e(w,s,\mathrm{y}(s)) ds \right) dw \bigg\| \\
\leq {}& d^{1-\varepsilon+\kappa\varepsilon-\kappa\xi} \bigg( \sup \big\| \mathcal{S}_{\kappa,\varepsilon}(\mathfrak{z}) \big[ \mathrm{y}_0 - \mathcal{N}(0,\mathrm{y}(0)) \big] \big\| + \big\| \mathcal{N}(\mathfrak{z},\mathrm{y}(\mathfrak{z})) \big\| \\
& + \sup \int_0^{\mathfrak{z}} (\mathfrak{z}-w)^{\kappa-1} \big\| \mathtt{A}^{1-\mu} \mathcal{Q}_\kappa(\mathfrak{z}-w) \big\| \big\| \mathtt{A}^\mu \mathcal{N}(w,\mathrm{y}(w)) \big\| dw \\
& + \sup \int_0^{\mathfrak{z}} (\mathfrak{z}-w)^{\kappa-1} \big\| \mathcal{Q}_\kappa(\mathfrak{z}-w) \big\| \left\| \mathcal{G}\left( w,\mathrm{y}(w),\int_0^w e(w,s,\mathrm{y}(w)) ds \right) \right\| dw \bigg) \\
\leq {}& d^{1-\varepsilon+\kappa\varepsilon-\kappa\xi} \left[ L'' d^{-1+\varepsilon-\kappa\varepsilon+\kappa\xi} (\mathrm{y}_0 - M_0 M_g) + M_0 M_g (1+P) \right] \\
& + d^{1-\varepsilon+\kappa\varepsilon-\kappa\xi} \left[ \left( \Lambda_{1-\mu} \frac{d^{\kappa\mu}\Gamma(1+\mu)}{\mu\Gamma(1+\kappa\mu)} (M_g(1+P)) \right) + L'\phi(\mathfrak{z})\Phi(\mathrm{y})[1+\psi(\mathfrak{z})] \frac{d^{\kappa\xi}}{\kappa\xi} \right] \\
\leq {}& \mathcal{M}_1^* + d^{\varepsilon(1-\kappa)-\kappa\xi-1} \left[ \left( \Lambda_{1-\mu} \frac{d^{\kappa\mu}\Gamma(1+\mu)}{\mu\Gamma(1+\kappa\mu)} (M_g(1+P)) \right) \right. \\
& \left. + L'\phi(\mathfrak{z})\Phi(\mathrm{y})[1+\psi(\mathfrak{z})] \frac{d^{\kappa\xi}}{\kappa\xi} \right].
\end{aligned}
$$

From Lemma 2 and hypotheses $(H_3)$, we have the boundness of the operators. Hence, it is bounded.

**Step 3:** Next, we show that the $z(\mathfrak{z})$ bounded maps are set to the equicontinuous set of $B_P(\mathcal{J})$.

Consider $0 < \mathfrak{z}_1 < \mathfrak{z}_2 \leq d$ and $\exists \, \mathcal{G} \in S_{\mathcal{G},\mathrm{y}}$, we have

$$\left\| z(\mathfrak{z}_2) - z(\mathfrak{z}_1) \right\|$$

$$\leq \left\| \mathfrak{z}_2^{1-\varepsilon+\kappa\varepsilon-\kappa\zeta} \left[ \mathcal{S}_{\kappa,\varepsilon}(\mathfrak{z}_2) \left[ \mathbf{y}_0 - \mathcal{N}(0, \mathbf{y}(0)) \right] + \mathcal{N}(\mathfrak{z}_2, \mathbf{y}(\mathfrak{z}_2)) \right. \right.$$

$$+ \int_0^{\mathfrak{z}_2} (\mathfrak{z}_2 - w)^{\kappa-1} \mathcal{Q}_\kappa(\mathfrak{z}_2 - w) \mathbf{A} \mathcal{N}(w, \mathbf{y}(w)) dw$$

$$\left. + \int_0^{\mathfrak{z}_2} (\mathfrak{z}_2 - w)^{\kappa-1} \mathcal{Q}_\kappa(\mathfrak{z}_2 - w) \mathcal{G}\left( w, \mathbf{y}(w), \int_0^w e(w, s, \mathbf{y}(s)) ds \right) dw \right]$$

$$- \mathfrak{z}_1^{1-\varepsilon+\kappa\varepsilon-\kappa\zeta} \left[ \mathcal{S}_{\kappa,\varepsilon}(\mathfrak{z}_1) \left[ \mathbf{y}_0 - \mathcal{N}(0, \mathbf{y}(0)) \right] + \mathcal{N}(\mathfrak{z}_1, \mathbf{y}(\mathfrak{z}_1)) \right.$$

$$+ \int_0^{\mathfrak{z}_1} (\mathfrak{z}_1 - w)^{\kappa-1} \mathcal{Q}_\kappa(\mathfrak{z}_1 - w) \mathbf{A} \mathcal{N}(w, \mathbf{y}(w)) dw$$

$$\left. \left. + \int_0^{\mathfrak{z}_1} (\mathfrak{z}_1 - w)^{\kappa-1} \mathcal{Q}_\kappa(\mathfrak{z}_1 - w) \mathcal{G}\left( w, \mathbf{y}(w), \int_0^w e(w, s, \mathbf{y}(s)) ds \right) dw \right] \right\|$$

$$\leq \left\| \left[ \mathfrak{z}_2^{1-\varepsilon+\kappa\varepsilon-\kappa\zeta} \mathcal{S}_{\kappa,\varepsilon}(\mathfrak{z}_2) - \mathfrak{z}_1^{1-\varepsilon+\kappa\varepsilon-\kappa\zeta} \mathcal{S}_{\kappa,\varepsilon}(\mathfrak{z}_1) \right] \left[ \mathbf{y}_0 - \mathcal{N}(0, \mathbf{y}(0)) \right] \right\|$$

$$+ \left\| \mathfrak{z}_2^{1-\varepsilon+\kappa\varepsilon-\kappa\zeta} \mathcal{N}(\mathfrak{z}_2, \mathbf{y}(\mathfrak{z}_2)) - \mathfrak{z}_1^{1-\varepsilon+\kappa\varepsilon-\kappa\zeta} \mathcal{N}(\mathfrak{z}_1, \mathbf{y}(\mathfrak{z}_1)) \right\|$$

$$+ \left\| \mathfrak{z}_2^{1-\varepsilon+\kappa\varepsilon-\kappa\zeta} \int_0^{\mathfrak{z}_1} (\mathfrak{z}_2 - w)^{\kappa-1} \mathcal{Q}_\kappa(\mathfrak{z}_2 - w) \mathbf{A} \mathcal{N}(w, \mathbf{y}(w)) dw \right.$$

$$+ \mathfrak{z}_2^{1-\varepsilon+\kappa\varepsilon-\kappa\zeta} \int_{\mathfrak{z}_1}^{\mathfrak{z}_2} (\mathfrak{z}_2 - w)^{\kappa-1} \mathcal{Q}_\kappa(\mathfrak{z}_2 - w) \mathbf{A} \mathcal{N}(w, \mathbf{y}(w)) dw$$

$$\left. - \mathfrak{z}_1^{1-\varepsilon+\kappa\varepsilon-\kappa\zeta} \int_0^{\mathfrak{z}_1} (\mathfrak{z}_1 - w)^{\kappa-1} \mathcal{Q}_\kappa(\mathfrak{z}_1 - w) \mathbf{A} \mathcal{N}(w, \mathbf{y}(w)) dw \right\|$$

$$+ \left\| \mathfrak{z}_2^{1-\varepsilon+\kappa\varepsilon-\kappa\zeta} \int_0^{\mathfrak{z}_1} (\mathfrak{z}_2 - w)^{\kappa-1} \mathcal{Q}_\kappa(\mathfrak{z}_2 - w) \mathcal{G}\left( w, \mathbf{y}(w), \int_0^w e(w, s, \mathbf{y}(s)) ds \right) dw \right.$$

$$+ \mathfrak{z}_2^{1-\varepsilon+\kappa\varepsilon-\kappa\zeta} \int_{\mathfrak{z}_1}^{\mathfrak{z}_2} (\mathfrak{z}_2 - w)^{\kappa-1} \mathcal{Q}_\kappa(\mathfrak{z}_2 - w) \mathcal{G}\left( w, \mathbf{y}(w), \int_0^w e(w, s, \mathbf{y}(s)) ds \right) dw$$

$$\left. - \mathfrak{z}_1^{1-\varepsilon+\kappa\varepsilon-\kappa\zeta} \int_0^{\mathfrak{z}_1} (\mathfrak{z}_1 - w)^{\kappa-1} \mathcal{Q}_\kappa(\mathfrak{z}_1 - w) \mathcal{G}\left( w, \mathbf{y}(w), \int_0^w e(w, s, \mathbf{y}(s)) ds \right) dw \right\|$$

$$\leq \left\| \left[ \mathfrak{z}_2^{1-\varepsilon+\kappa\varepsilon-\kappa\zeta} \mathcal{S}_{\kappa,\varepsilon}(\mathfrak{z}_2) - \mathfrak{z}_1^{1-\varepsilon+\kappa\varepsilon-\kappa\zeta} \mathcal{S}_{\kappa,\varepsilon}(\mathfrak{z}_1) \right] \left[ \mathbf{y}_0 - \mathcal{N}(0, \mathbf{y}(0)) \right] \right\|$$

$$+ \left\| \mathfrak{z}_2^{1-\varepsilon+\kappa\varepsilon-\kappa\zeta} \mathcal{N}(\mathfrak{z}_2, \mathbf{y}(\mathfrak{z}_2)) - \mathfrak{z}_1^{1-\varepsilon+\kappa\varepsilon-\kappa\zeta} \mathcal{N}(\mathfrak{z}_1, \mathbf{y}(\mathfrak{z}_1)) \right\|$$

$$+ \left\| \mathfrak{z}_2^{1-\varepsilon+\kappa\varepsilon-\kappa\zeta} \int_{\mathfrak{z}_1}^{\mathfrak{z}_2} (\mathfrak{z}_2 - w)^{\kappa-1} \mathcal{Q}_\kappa(\mathfrak{z}_2 - w) \mathcal{N}(w, \mathbf{y}(w)) dw \right\|$$

$$+ \left\| \mathfrak{z}_2^{1-\varepsilon+\kappa\varepsilon-\kappa\zeta} \int_0^{\mathfrak{z}_1} (\mathfrak{z}_2 - w)^{\kappa-1} \mathcal{Q}_\kappa(\mathfrak{z}_2 - w) \mathbf{A} \mathcal{N}(w, \mathbf{y}(w)) dw \right.$$

$$\left. - \mathfrak{z}_1^{1-\varepsilon+\kappa\varepsilon-\kappa\zeta} \int_0^{\mathfrak{z}_1} (\mathfrak{z}_1 - w)^{\kappa-1} \mathcal{Q}_\kappa(\mathfrak{z}_2 - w) \mathbf{A} \mathcal{N}(w, \mathbf{y}(w)) dw \right\|$$

$$+ \left\| \mathfrak{z}_1^{1-\varepsilon+\kappa\varepsilon-\kappa\zeta} \int_0^{\mathfrak{z}_1} (\mathfrak{z}_1 - w)^{\kappa-1} \mathcal{Q}_\kappa(\mathfrak{z}_2 - w) \mathbf{A} \mathcal{N}(w, \mathbf{y}(w)) dw \right.$$

$$\left. - \mathfrak{z}_1^{1-\varepsilon+\kappa\varepsilon-\kappa\zeta} \int_0^{\mathfrak{z}_1} (\mathfrak{z}_1 - w)^{\kappa-1} \mathcal{Q}_\kappa(\mathfrak{z}_1 - w) \mathbf{A} \mathcal{N}(w, \mathbf{y}(w)) dw \right\|$$

$$+ \left\| \mathfrak{z}_2^{1-\varepsilon+\kappa\varepsilon-\kappa\zeta} \int_{\mathfrak{z}_1}^{\mathfrak{z}_2} (\mathfrak{z}_2 - w)^{\kappa-1} \mathcal{Q}_\kappa(\mathfrak{z}_2 - w) \mathcal{G}\left( w, \mathbf{y}(w), \int_0^w e(w, s, \mathbf{y}(s)) ds \right) dw \right\|$$

$$+ \left\| \mathfrak{z}_2^{1-\varepsilon+\kappa\varepsilon-\kappa\zeta} \int_0^{\mathfrak{z}_1} (\mathfrak{z}_2 - w)^{\kappa-1} \mathcal{Q}_\kappa(\mathfrak{z}_2 - w) \mathcal{G}\left( w, \mathbf{y}(w), \int_0^w e(w, s, \mathbf{y}(s)) ds \right) dw \right.$$

$$\left. - \mathfrak{z}_1^{1-\varepsilon+\kappa\varepsilon-\kappa\zeta} \int_0^{\mathfrak{z}_1} (\mathfrak{z}_1 - w)^{\kappa-1} \mathcal{Q}_\kappa(\mathfrak{z}_2 - w) \mathcal{G}\left( w, \mathbf{y}(w), \int_0^w e(w, s, \mathbf{y}(s)) ds \right) dw \right\|$$

$$+ \left\| \mathfrak{z}_1^{1-\varepsilon+\kappa\varepsilon-\kappa\zeta} \int_0^{\mathfrak{z}_1} (\mathfrak{z}_1 - w)^{\kappa-1} \mathcal{Q}_\kappa(\mathfrak{z}_2 - w) \mathcal{G}\left( w, \mathbf{y}(w), \int_0^w e(w, s, \mathbf{y}(s)) ds \right) dw \right.$$

$$\left. - \mathfrak{z}_1^{1-\varepsilon+\kappa\varepsilon-\kappa\zeta} \int_0^{\mathfrak{z}_1} (\mathfrak{z}_1 - w)^{\kappa-1} \mathcal{Q}_\kappa(\mathfrak{z}_1 - w) \mathcal{G}\left( w, \mathbf{y}(w), \int_0^w e(w, s, \mathbf{y}(s)) ds \right) dw \right\|$$

$$= \sum_{i=1}^8 I_i.$$

Since $\mathcal{S}_{\kappa,\varepsilon}(\mathfrak{z})(\mathrm{y}_0 - M_0 M_g)$ is strong-continuous, we have

$$I_1 \text{ tends to 0 as } \mathfrak{z}_2 \to \mathfrak{z}_1.$$

The equicontinuity of $\mathcal{N}$ ensures that

$$I_2 \text{ tends to 0, as } \mathfrak{z}_2 \to \mathfrak{z}_1.$$

$$I_3 = \left\| \mathfrak{z}_2^{1-\varepsilon+\kappa\varepsilon-\kappa\xi} \int_{\mathfrak{z}_1}^{\mathfrak{z}_2} (\mathfrak{z}_2 - w)^{\kappa-1} \mathcal{Q}_\kappa(\mathfrak{z}_2 - w) \mathtt{A} \mathcal{N}(w, \mathrm{y}(w)) dw \right\|$$

$$\leq \mathfrak{z}_2^{1-\varepsilon+\kappa\varepsilon-\kappa\xi} \Lambda_{1-\mu} M_g(1+P) \frac{\Gamma(1+\mu)}{\mu\Gamma(1+\kappa\mu)} (\mathfrak{z}_2 - \mathfrak{z}_1)^{\kappa\mu}$$

Then, $I_3$ tends 0 as $\mathfrak{z}_2 \to \mathfrak{z}_1$.

$$I_4 = \left\| \mathfrak{z}_2^{1-\varepsilon+\kappa\varepsilon-\kappa\xi} \int_0^{\mathfrak{z}_1} (\mathfrak{z}_2 - w)^{\kappa-1} \mathcal{Q}_\kappa(\mathfrak{z}_2 - w) \mathtt{A} \mathcal{N}(w, \mathrm{y}(w)) dw \right.$$

$$\left. - \mathfrak{z}_1^{1-\varepsilon+\kappa\varepsilon-\kappa\xi} \int_0^{\mathfrak{z}_1} (\mathfrak{z}_1 - w)^{\kappa-1} \mathcal{Q}_\kappa(\mathfrak{z}_2 - w) \mathtt{A} \mathcal{N}(w, \mathrm{y}(w)) dw \right\|$$

$$\leq \kappa \Lambda_{1-\mu} M_g(1+P) \frac{\Gamma(1+\mu)}{\mu\Gamma(1+\kappa\mu)}$$

$$\times \left\| \int_0^{\mathfrak{z}_1} \left( \mathfrak{z}_2^{1-\varepsilon+\kappa\varepsilon-\kappa\xi} (\mathfrak{z}_2 - w)^{\kappa-1} - \mathfrak{z}_1^{1-\varepsilon+\kappa\varepsilon-\kappa\xi} (\mathfrak{z}_1 - w)^{\kappa-1} \right) (\mathfrak{z}_2 - w)^{\kappa(\mu-1)} dw \right\|.$$

We have, $I_4$ tends 0 as $\mathfrak{z}_2 \to \mathfrak{z}_1$. Also,

$$I_5 = \left\| \mathfrak{z}_1^{1-\varepsilon+\kappa\varepsilon-\kappa\xi} \int_0^{\mathfrak{z}_1} \left( (\mathfrak{z}_1 - w)^{\kappa-1} \mathcal{Q}_\kappa(\mathfrak{z}_2 - w) \mathtt{A} \mathcal{N}(w, \mathrm{y}(w)) \right. \right.$$

$$\left. \left. - (\mathfrak{z}_1 - w)^{\kappa-1} \mathcal{Q}_\kappa(\mathfrak{z}_1 - w) \mathtt{A} \mathcal{N}(w, \mathrm{y}(w)) \right) dw \right\|$$

$$\leq M_0' M_g(1+P) \mathfrak{z}_1^{1-\varepsilon+\kappa\varepsilon-\kappa\xi} \int_0^{\mathfrak{z}_1} (\mathfrak{z}_1 - w)^{\kappa-1} \left\| [\mathcal{Q}_\kappa(\mathfrak{z}_2 - w) - \mathcal{Q}_\kappa(\mathfrak{z}_1 - w)] \right\|.$$

By Theorem 1 and strong continuity of $\mathcal{Q}_\kappa(\mathfrak{z})$, $I_5$ tends to 0, as $\mathfrak{z}_2 \to \mathfrak{z}_1$.

$$I_6 = \left\| \mathfrak{z}_2^{1-\varepsilon+\kappa\varepsilon-\kappa\xi} \int_{\mathfrak{z}_1}^{\mathfrak{z}_2} (\mathfrak{z}_2 - w)^{\kappa-1} \mathcal{Q}_\kappa(\mathfrak{z}_2 - w) \mathcal{G}\left( w, \mathrm{y}(w), \int_0^w e(w, s, \mathrm{y}(s)) ds \right) dw \right\|$$

$$\leq L' \left| \mathfrak{z}_2^{1-\varepsilon+\kappa\varepsilon-\kappa\xi} \int_{\mathfrak{z}_1}^{\mathfrak{z}_2} (\mathfrak{z}_2 - w)^{\kappa\xi-1} \phi(w) \Phi(\mathrm{y}) [1 + \psi(\mathfrak{z})] dw \right|$$

$$\leq L' \int_0^{\mathfrak{z}_1} \left[ \mathfrak{z}_1^{1-\varepsilon+\kappa\varepsilon-\kappa\xi} (\mathfrak{z}_1 - w)^{\kappa\xi-1} - \mathfrak{z}_2^{(1+\kappa\xi)(1-\kappa)} (\mathfrak{z}_2 - w)^{\kappa\xi-1} \right]$$

$$\times \phi(w) \Phi(\mathrm{y}) [1 + \psi(\mathfrak{z})] dw.$$

Then $I_6$ *tends to 0 as* $\mathfrak{z}_2 \to \mathfrak{z}_1$ by using $(H_2)$ and the Lebesgue-dominated convergent theorem.

$$I_7 = \left\| \mathfrak{z}_2^{1-\varepsilon+\kappa\varepsilon-\kappa\xi} \int_0^{\mathfrak{z}_1} (\mathfrak{z}_2 - w)^{\kappa-1} \mathcal{Q}_\kappa(\mathfrak{z}_2 - w) \mathcal{G}\left( w, \mathrm{y}(w), \int_0^w e(w, s, \mathrm{y}(s)) ds \right) dw \right.$$

$$\left. - \mathfrak{z}_1^{1-\varepsilon+\kappa\varepsilon-\kappa\xi} \int_0^{\mathfrak{z}_1} (\mathfrak{z}_1 - w)^{\kappa-1} \mathcal{Q}_\kappa(\mathfrak{z}_2 - w) \mathcal{G}\left( w, \mathrm{y}(w), \int_0^w e(w, s, \mathrm{y}(s)) ds \right) dw \right\|$$

$$\leq L' \int_0^{\mathfrak{z}_1} (\mathfrak{z}_2 - w)^{-\kappa+\kappa\xi} \left| \mathfrak{z}_2^{1-\varepsilon+\kappa\varepsilon-\kappa\xi} (\mathfrak{z}_2 - w)^{\kappa-1} - \mathfrak{z}_1^{1-\varepsilon+\kappa\varepsilon-\kappa\xi} (\mathfrak{z}_1 - w)^{\kappa-1} \right|$$

$$\times \phi(w) \Phi(\mathrm{y}) [1 + \psi(\mathfrak{z})] dw,$$

and $\int_0^{\mathfrak{z}_1} 2\mathfrak{z}_1^{(1+\kappa\xi)(1-\kappa)}(\mathfrak{z}_1 - w)^{\kappa\xi-1}\phi(w)\Phi(y)[1 + \psi(\mathfrak{z})]dw$ exists $(w \in (0, \mathfrak{z}_1])$, then from Lebesgue's dominated convergence theorem, we obtain

$$\int_0^{\mathfrak{z}_1} (\mathfrak{z}_2 - w)^{-\kappa+\kappa\xi}\left|\mathfrak{z}_2^{1-\varepsilon+\kappa\varepsilon-\kappa\xi}(\mathfrak{z}_2 - w)^{\kappa-1} - \mathfrak{z}_1^{1-\varepsilon+\kappa\varepsilon-\kappa\xi}(\mathfrak{z}_1 - w)^{\kappa-1}\right|\phi(w)\Phi(y)[1 + \psi(\mathfrak{z})]dw$$

$$\to 0 \text{ as } \mathfrak{z}_2 \to \mathfrak{z}_1,$$

so we conclude $\lim_{\mathfrak{z}_2 \to \mathfrak{z}_1} I_7 = 0$.

For any $\epsilon > 0$, we have

$$I_8 = \left\|\int_0^{\mathfrak{z}_1} \mathfrak{z}_1^{1-\varepsilon+\kappa\varepsilon-\kappa\xi}\left[\mathcal{Q}_\kappa(\mathfrak{z}_2 - w) - \mathcal{Q}_\kappa(\mathfrak{z}_1 - w)\right](\mathfrak{z}_1 - w)^{\kappa-1}\mathcal{G}\left(w, y(w), \int_0^w e(w, s, y(s))ds\right)dw\right\|$$

$$\leq \mathfrak{z}_1^{1-\varepsilon+\kappa\varepsilon-\kappa\xi+\kappa(1+\xi)}\int_0^{\mathfrak{z}_1}(\mathfrak{z}_1 - w)^{\kappa\xi-1}\phi(w)\Phi(y)[1 + \psi(\mathfrak{z})]dw$$

$$\times \sup_{w\in[0,\mathfrak{z}_1-\epsilon]}\left\|\mathcal{Q}_\kappa(\mathfrak{z}_2 - w) - \mathcal{Q}_\kappa(\mathfrak{z}_1 - w)\right\|$$

$$+ 2L'\int_{\mathfrak{z}_1-\epsilon}^{\mathfrak{z}_1}\mathfrak{z}_1^{1-\varepsilon+\kappa\varepsilon+\kappa\xi}(\mathfrak{z}_1 - w)^{\kappa\xi-1}\phi(w)\Phi(y)[1 + \psi(\mathfrak{z})]dw.$$

From Theorem (1) and $\lim_{\mathfrak{z}_2 \to \mathfrak{z}_1} I_6 = 0$, we have $I_8 \to 0$ independently of $y \in B_P(\mathcal{J})$ as $\mathfrak{z}_2 \to \mathfrak{z}_1, \epsilon \to 0$. Hence, $\|z(\mathfrak{z}_2) - z(\mathfrak{z}_1)\| \to 0$ independently of $y \in B_P(\mathcal{J})$ as $\mathfrak{z}_2 \to \mathfrak{z}_1$. Therefore, $\{\Psi y(\mathfrak{z}) : y \in B_P(\mathcal{J})\}$ is equicontinuous on $\mathcal{J}$.

**Step 4:** Show the relative compact of $V(\mathfrak{z}) = \{z(\mathfrak{z}) : z \in \Psi(B_P(\mathcal{J}))\}$ for $\mathfrak{z} \in \mathcal{J}$.

Let $0 < \alpha < \mathfrak{z}$, and there is a positive value $q$, assume an operator $z'(\mathfrak{z})$ on $B_P(\mathcal{J})$ by

$$z'_{\alpha,q}(\mathfrak{z}) = \mathfrak{z}^{1-\varepsilon+\kappa\varepsilon-\kappa\xi}\left[\mathcal{S}_{\kappa,\varepsilon}(\mathfrak{z})[y_0 - \mathcal{N}(0, y(0))] + \mathcal{N}(\mathfrak{z}, y(\mathfrak{z}))\right.$$

$$+ \int_0^{\mathfrak{z}-\alpha}(\mathfrak{z} - w)^{\kappa-1}\mathcal{Q}_\kappa(\mathfrak{z} - w)\mathbf{A}\mathcal{N}(w, y(w))dw$$

$$\left.+ \int_0^{\mathfrak{z}-\alpha}(\mathfrak{z} - w)^{\kappa-1}\mathcal{Q}_\kappa(\mathfrak{z} - w)\mathcal{G}\left(w, y(w), \int_0^w e(w, s, y(s))ds\right)dw\right]$$

$$= \mathfrak{z}^{1-\varepsilon+\kappa\varepsilon-\kappa\xi}\left[\mathcal{S}_{\kappa,\varepsilon}(\mathfrak{z})[y_0 - \mathcal{N}(0, y(0))] + \mathcal{N}(\mathfrak{z}, y(\mathfrak{z}))\right.$$

$$+ \int_0^{\mathfrak{z}-\alpha}\int_q^\infty \kappa\theta M_\kappa(\theta)(\mathfrak{z} - w)^{\kappa-1}T((\mathfrak{z} - w)^\kappa\theta)\mathbf{A}\mathcal{N}(w, y(w))dw$$

$$\left.+ \int_0^{\mathfrak{z}-\alpha}\int_q^\infty \kappa\theta M_\kappa(\theta)(\mathfrak{z} - w)^{\kappa-1}T((\mathfrak{z} - w)^\kappa\theta)\mathcal{G}\left(w, y(w), \int_0^w e(w, s, y(s))ds\right)d\theta dw\right]$$

$$= \mathfrak{z}^{1-\varepsilon+\kappa\varepsilon-\kappa\xi}\left[\mathcal{S}_{\kappa,\varepsilon}(\mathfrak{z})[y_0 - \mathcal{N}(0, y(0))] + \mathcal{N}(\mathfrak{z}, y(\mathfrak{z}))\right]$$

$$+ \kappa\mathfrak{z}^{1-\varepsilon+\kappa\varepsilon-\kappa\xi}T(\alpha^\kappa q)\int_0^{\mathfrak{z}-q}\int_q^\infty \theta M_\kappa(\theta)(\mathfrak{z} - w)^{\kappa-1}$$

$$\times T((\mathfrak{z} - w)^\kappa\theta - \alpha^\kappa q)\left[\mathbf{A}\mathcal{N}(w, y(w)) + \mathcal{G}\left(w, y(w), \int_0^w e(w, s, y(s))\right)\right]d\theta dw.$$

From the compactness of $T(\alpha^\kappa q)$, we note that $V_{\alpha,\xi}(\mathfrak{z}) = \{(z'_{\alpha,q}(\mathfrak{z}))\mathbf{y}(\mathfrak{z}) : \mathbf{y} \in B_P(\mathcal{J})\}$ is pre-compact in Y. $\forall \, \mathbf{y} \in B_P(\mathcal{J})$, we have

$$\left\| z(\mathfrak{z}) - z'_{\alpha,q}(\mathfrak{z}) \right\|$$

$$\leq \left\| \kappa_{\mathfrak{z}}^{1-\varepsilon+\kappa\varepsilon-\kappa\xi} \int_0^{\mathfrak{z}} \int_0^q \theta M_\kappa(\theta)(\mathfrak{z}-w)^{\kappa-1} T((\mathfrak{z}-w)^\kappa \theta) \right.$$

$$\left. \left[ \mathtt{A}\mathcal{N}(w,\mathbf{y}(w)) + \mathcal{G}\left(w,\mathbf{y}(w),\int_0^w e(w,s,\mathbf{y}(s))ds\right)\right] d\theta dw \right\|$$

$$+ \left\| \kappa_{\mathfrak{z}}^{1-\varepsilon+\kappa\varepsilon-\kappa\xi} \int_{\mathfrak{z}-\alpha}^{\mathfrak{z}} \int_q^\infty (\mathfrak{z}-w)^{\kappa-1} \theta M_\kappa(\theta) T((\mathfrak{z}-w)^\kappa \theta) \right.$$

$$\left. \left[ \mathtt{A}\mathcal{N}(w,\mathbf{y}(w)) + \mathcal{G}\left(w,\mathbf{y}(w),\int_0^w e(w,s,\mathbf{y}(s))ds\right)\right] d\theta dw \right\|$$

$$\leq \kappa \Lambda_0 \mathfrak{z}^{1-\varepsilon+\kappa\varepsilon-\kappa\xi} \left( \int_0^{\mathfrak{z}} \int_0^q \theta M_\kappa(\theta)(\mathfrak{z}-w)^{\kappa-1}(\mathfrak{z}-w)^{\kappa\xi-\kappa}\theta^{\xi-1} \right.$$

$$\times \left[ M'_0 M_g(1+P) + \phi(w)\Phi(\mathbf{y})[1+\psi(\mathfrak{z})] \right] d\theta dw$$

$$+ \left. \int_{\mathfrak{z}-\alpha}^{\mathfrak{z}} \int_q^\infty (\mathfrak{z}-w)^{\kappa-1} \theta M_\kappa(\theta)(\mathfrak{z}-w)^{\kappa\xi-\kappa}\theta^{\xi-1} \left[ M'_0 M_g(1+P) + \phi(w)\Phi(\mathbf{y})[1+\psi(\mathfrak{z})] \right] dw \right)$$

$$\leq \kappa \Lambda_0 \mathfrak{z}^{1-\varepsilon+\kappa\varepsilon-\kappa\xi} \left( \int_0^{\mathfrak{z}} (\mathfrak{z}-w)^{\kappa\xi-1} \left[ M'_0 M_g(1+P) + \phi(w)\Phi(\mathbf{y})[1+\psi(\mathfrak{z})] \right] dw \int_0^q \theta^\xi M_\kappa(\theta)d\theta \right.$$

$$+ \left. \int_{\mathfrak{z}-\alpha}^{\mathfrak{z}} (\mathfrak{z}-w)^{\kappa\xi-1} \left[ M'_0 M_g(1+P) + \phi(w)\Phi(\mathbf{y})[1+\psi(\mathfrak{z})] \right] dw \int_0^\infty \theta^\xi M_\kappa(\theta)d\theta \right)$$

$$\leq \kappa \Lambda_0 \mathfrak{z}^{1-\varepsilon+\kappa\varepsilon-\kappa\xi} \left( \int_0^{\mathfrak{z}} (\mathfrak{z}-w)^{\kappa\xi-1} \left[ M'_0 M_g(1+P) + \phi(w)\Phi(\mathbf{y})[1+\psi(\mathfrak{z})] \right] dw \int_0^q \theta^\xi M_\kappa(\theta)d\theta \right.$$

$$+ \left. \frac{\Gamma(1-\xi)}{\Gamma(1-\kappa\xi)} \int_{\mathfrak{z}-\alpha}^{\mathfrak{z}} (\mathfrak{z}-w)^{\kappa\xi-1} \left[ M'_0 M_g(1+P) + \phi(w)\Phi(\mathbf{y})[1+\psi(\mathfrak{z})] \right] dw \right)$$

$\to 0$ as $\alpha$ *tends to* $0$, $q$ *tends to* $0$.

So, $V_{\alpha,q}(\mathfrak{z}) = \{z_{\alpha,q}(\mathfrak{z}) : \mathfrak{z} \in B_P(\mathcal{J})\}$ are arbitrary closed to $V(\mathfrak{z}) = \{z(\mathfrak{z}) : \mathfrak{z} \in B_P(\mathcal{I})\}$. Therefore, $\{z(\mathfrak{z}) : \mathfrak{z} \in B_P(\mathcal{J})\}$ is relatively compact by the Arzela–Ascoli theorem. Thus, the continuity of $z(\mathfrak{z})$ and relative compactness of $\{z(\mathfrak{z}) : \mathfrak{z} \in B_P(\mathcal{J})\}$ imply that $z(\mathfrak{z})$ is a completely continuous operator.

**Step 5:** $\Psi$ has a closed graph.

Take $\mathbf{y}_n \to \mathbf{y}_*$ as $n \to \infty$, $z_n(\mathfrak{z}) \in \Psi(\mathbf{y}_n)$ and $z_n \to z_*$ as $n \to \infty$, we have to show that $z_* \in \Psi(\mathbf{y}_*)$. Since $z_n \in \Psi(\mathbf{y}_n)$ then $\exists$ a function $\mathcal{G}_n \in S_{\mathcal{G},\mathbf{y}_n}$, such that

$$z_n(\mathfrak{z}) = \mathfrak{z}^{1-\varepsilon+\kappa\varepsilon-\kappa\xi} \left[ \mathcal{S}_{\kappa,\varepsilon}(\mathfrak{z})[\mathbf{y}_0 - \mathcal{N}(0,\mathbf{y}(0))] + \mathcal{N}(\mathfrak{z},\mathbf{y}_n(\mathfrak{z})) \right.$$

$$+ \left. \int_0^{\mathfrak{z}} (\mathfrak{z}-w)^{\kappa-1} \mathcal{Q}_\kappa(\mathfrak{z}-w) \mathtt{A}\mathcal{N}(w,\mathbf{y}_n(w))dw + \int_0^{\mathfrak{z}} (\mathfrak{z}-w)^{\kappa-1} \mathcal{Q}_\kappa(\mathfrak{z}-w) \mathcal{G}_n(w)dw \right].$$

We need to show that $\exists \, \mathcal{G}_* \in S_{\mathcal{G},\mathbf{y}_*}$, such that

$$z_*(\mathfrak{z}) = \mathfrak{z}^{1-\varepsilon+\kappa\varepsilon-\kappa\xi} \left[ \mathcal{S}_{\kappa,\varepsilon}(\mathfrak{z})[\mathbf{y}_0 - \mathcal{N}(0,\mathbf{y}(0))] + \mathcal{N}(\mathfrak{z},\mathbf{y}_*(\mathfrak{z})) \right.$$

$$+ \left. \int_0^{\mathfrak{z}} (\mathfrak{z}-w)^{\kappa-1} \mathcal{Q}_\kappa(\mathfrak{z}-w) \mathtt{A}\mathcal{N}(w,\mathbf{y}_*(w))dw + \int_0^{\mathfrak{z}} (\mathfrak{z}-w)^{\kappa-1} \mathcal{Q}_\kappa(\mathfrak{z}-w) \mathcal{G}_*(w)dw \right].$$

Clearly,

$$\left\| \left[ z_n(\mathfrak{z}) - \mathfrak{z}^{1-\varepsilon+\kappa\varepsilon-\kappa\zeta} \left( \mathcal{S}_{\kappa,\varepsilon}(\mathfrak{z}) \left[ \mathbf{y}_0 + \mathcal{N}(0, \mathbf{y}(0)) \right] - \mathcal{N}(\mathfrak{z}, \mathbf{y}_n(\mathfrak{z})) \right.\right.\right.$$

$$\left. - \int_0^{\mathfrak{z}} (\mathfrak{z} - w)^{\mathfrak{z}-1} \mathcal{Q}_\kappa(\mathfrak{z} - w) \mathtt{A}\mathcal{N}(w, \mathbf{y}_n(w)) dw \right) \right]$$

$$- \left[ z_*(\mathfrak{z}) - \mathfrak{z}^{1-\varepsilon+\kappa\varepsilon-\kappa\zeta} \left( \mathcal{S}_{\kappa,\varepsilon}(\mathfrak{z}) \left[ \mathbf{y}_0 - \mathcal{N}(0, \mathbf{y}(0)) \right] - \mathcal{N}(\mathfrak{z}, \mathbf{y}_*(\mathfrak{z})) \right.\right.$$

$$\left.\left.\left. - \int_0^{\mathfrak{z}} (\mathfrak{z} - w)^{\mathfrak{z}-1} \mathcal{Q}_\kappa(\mathfrak{z} - w) \mathtt{A}\mathcal{N}(w, \mathbf{y}_*(w)) dw \right) \right] \right\| \to 0 \text{ as } n \to \infty.$$

Next, we define the operator $Y : L'(\mathcal{J}, Y) \to \mathcal{X}$,

$$Y(g)(\mathfrak{z}) = \int_0^{\mathfrak{z}} (\mathfrak{z} - w)^{\kappa-1} \mathcal{Q}_\kappa(\mathfrak{z} - w) \mathcal{G}\left( w, \mathbf{y}(w), \int_0^w e(w, s, \mathbf{y}(w)) ds \right) dw.$$

We have (by (5)) that $Y \circ S_{\mathcal{G},\mathbf{y}}$ is a closed graph operator. So, by referring to y*psilon*, we know

$$\left[ z_n(\mathfrak{z}) - \mathfrak{z}^{1-\varepsilon+\kappa\varepsilon-\kappa\zeta} \left( \mathcal{S}_{\kappa,\varepsilon}(\mathfrak{z}) \left[ \mathbf{y}_0 + \mathcal{N}(0, \mathbf{y}(0)) \right] - \mathcal{N}(\mathfrak{z}, \mathbf{y}_n(\mathfrak{z})) \right.\right.$$

$$\left.\left. - \int_0^{\mathfrak{z}} (\mathfrak{z} - w)^{\mathfrak{z}-1} \mathcal{Q}_\kappa(\mathfrak{z} - w) \mathtt{A}\mathcal{N}(w, \mathbf{y}_n(w)) dw \right) \right] \in Y(S_{\mathcal{G},\mathbf{y}_n}),$$

since $\mathcal{G}_n \to \mathcal{G}_*$, we follow from (5) that

$$\left[ z_*(\mathfrak{z}) - \mathfrak{z}^{1-\varepsilon+\kappa\varepsilon-\kappa\zeta} \left( \mathcal{S}_{\kappa,\varepsilon}(\mathfrak{z}) \left[ \mathbf{y}_0 - \mathcal{N}(0, \mathbf{y}(0)) \right] - \mathcal{N}(\mathfrak{z}, \mathbf{y}_*(\mathfrak{z})) \right.\right.$$

$$\left.\left. - \int_0^{\mathfrak{z}} (\mathfrak{z} - w)^{\mathfrak{z}-1} \mathcal{Q}_\kappa(\mathfrak{z} - w) \mathtt{A}\mathcal{N}(w, \mathbf{y}_*(w)) dw \right) \right] \in Y(S_{\mathcal{G},u_*}).$$

Therefore, $\Psi$ is a closed graph.
**Step:6** Set $\Lambda$ is bounded.

$$\Lambda = \{ \mathbf{y} \in \partial B_P(\mathcal{J}) : \lambda \mathbf{y} = \Psi(\mathbf{y}) \text{ for some } \lambda > 1 \}.$$

Let $\mathbf{y} \in \Lambda$. Then $\lambda w \in \Psi(\mathbf{y})$ for some $\lambda > 1$. Thus, there exists $\mathcal{G} \in S_{\mathcal{G},\mathbf{y}}$ in ways that for each $\mathfrak{z} \in [0, d]$ and $\|\mathtt{A}^{1-\mu}\| \le M_0'$, we have

$$\mathbf{y}(\mathfrak{z}) = \lambda^{-1} \mathfrak{z}^{1-\varepsilon+\kappa\varepsilon-\kappa\zeta} \left[ \mathcal{S}_{\kappa,\varepsilon}(\mathfrak{z}) \left[ \mathbf{y}_0 - \mathcal{N}(0, \mathbf{y}(0)) \right] + \mathcal{N}(\mathfrak{z}, \mathbf{y}(\mathfrak{z})) \right.$$

$$+ \int_0^{\mathfrak{z}} (\mathfrak{z} - w)^{\kappa-1} \mathcal{Q}_\kappa(\mathfrak{z} - w) \mathtt{A}\mathcal{N}(w, \mathbf{y}(w)) dw$$

$$\left. + \int_0^{\mathfrak{z}} (\mathfrak{z} - w)^{\kappa-1} \mathcal{Q}_\kappa(\mathfrak{z} - w) \mathcal{G}\left( w, \mathbf{y}(w), \int_0^w e(w, s, \mathbf{y}(s)) ds \right) \right] dw.$$

By assumptions $(H_2) - (H_4)$, we have

$$\|y(\mathfrak{z})\| = \left\| \lambda^{-1} \mathfrak{z}^{1-\varepsilon+\kappa\varepsilon-\kappa\xi} \left[ \mathcal{S}_{\kappa,\varepsilon}(\mathfrak{z}) \left[ y_0 - \mathcal{N}(0, y(0)) \right] + \mathcal{N}(\mathfrak{z}, y(\mathfrak{z})) \right. \right.$$

$$+ \int_0^{\mathfrak{z}} (\mathfrak{z} - w)^{\kappa-1} \mathcal{Q}_\kappa(\mathfrak{z} - w) \mathsf{A} \mathcal{N}(w, y(w)) dw$$

$$\left. \left. + \int_0^{\mathfrak{z}} (\mathfrak{z} - w)^{\kappa-1} \mathcal{Q}_\kappa(\mathfrak{z} - w) \mathcal{G}\left( w, y(w), \int_0^w e(w, s, y(s)) ds \right) \right] dw \right\|$$

$$\leq d^{1-\varepsilon+\kappa\varepsilon-\kappa\xi} \left[ \sup \left\| \mathcal{S}_{\kappa,\varepsilon}(\mathfrak{z}) \left[ y_0 - \mathcal{N}(0, y(0)) \right] \right\| + \left\| \mathcal{N}(\mathfrak{z}, y(\mathfrak{z})) \right\| \right.$$

$$\left. + \sup \int_0^{\mathfrak{z}} (\mathfrak{z} - w)^{\kappa-1} \left\| \mathcal{Q}_\kappa(\mathfrak{z} - w) \right\| \left( \left\| \mathsf{A} \mathcal{N}(w, y(w)) \right\| + \left\| \mathcal{G}\left( w, y(w), \int_0^w e(w, s, y(s)) ds \right) \right\| \right) dw \right]$$

$$\leq d^{1-\varepsilon+\kappa\varepsilon-\kappa\xi} \left[ L'' d^{-1+\varepsilon-\kappa\varepsilon+\kappa\xi} (y_0 - M_0 M_g) + M_0 M_g(1 + P) \right]$$

$$+ d^{1-\varepsilon+\kappa\varepsilon-\kappa\xi} L' \int_0^{\mathfrak{z}} (\mathfrak{z} - w)^{\kappa\xi-1} \left[ M_0' M_g(1 + P) + \phi(w) \Phi(\|y(w)\|)(1 + \psi(w)) \right] dw$$

$$\leq M_1^* + L' M_2^* + d^{1-\varepsilon+\kappa\varepsilon-\kappa\xi} L' \int_0^{\mathfrak{z}} (\mathfrak{z} - w)^{\kappa\xi-1} \phi(w) \Phi(\|y(w)\|)(1 + \psi(w)) dw,$$

$$\text{where } M_1^* = d^{1-\varepsilon+\kappa\varepsilon-\kappa\xi} \left[ L'' d^{-1+\varepsilon-\kappa\varepsilon+\kappa\xi} (y_0 - M_0 M_g) + M_0 M_g(1 + P) \right]$$

$$\text{and } M_2^* = d^{1-\varepsilon(1+\kappa\xi)} \frac{M_0' M_g(1 + P)}{\kappa\xi}.$$

Consider the RHS of the above inequality as $\gamma(\mathfrak{z})$. Then, we have

$$\gamma(0) = M_1^*, \ \|y(\mathfrak{z})\| \leq \gamma(\mathfrak{z}), \ \mathfrak{z} \in [0, d],$$
$$\gamma'(\mathfrak{z}) = d^{1-\varepsilon+\kappa\varepsilon-\kappa\xi} L' (w - \mathfrak{z})^{\kappa\xi-1} \phi(\mathfrak{z}) \Phi(\|y(\mathfrak{z})\|)(1 + \psi(\mathfrak{z})).$$

By the non-decreasing character of $\Phi$, we obtain

$$\gamma'(\mathfrak{z}) = d^{1-\varepsilon+\kappa\varepsilon-\kappa\xi} L' (w - \mathfrak{z})^{\kappa\xi-1} \phi(\mathfrak{z}) \Phi(\gamma(\mathfrak{z}))(1 + \psi(\mathfrak{z})).$$

Then the above inequality implies (for each $\mathfrak{z} \in \mathcal{J}$) that

$$\int_{\gamma(0)}^{\gamma(\mathfrak{z})} \frac{du}{\Phi(u)} \leq L' \int_0^{\mathfrak{z}} (\mathfrak{z} - w)^{\kappa\xi-1} \phi(\mathfrak{z})(1 + \psi(\mathfrak{z})) dw < \int_{M_1^*}^{\infty} \frac{du}{\Phi(u)}.$$

This inequality implies that there exists a constant $\mathcal{L}$, such that $\gamma(\mathfrak{z}) \leq \mathcal{L}$, $\mathfrak{z} \in \mathcal{J}$, and, hence, $y(\mathfrak{z}) \leq \mathcal{L}$. From this we notice that set $\Lambda$ is bounded. Therefore, by [17], Martelli's fixed point theorem $\Psi$ has a fixed point, which is the mild solution of the system (1)–(2). $\square$

## 4. Example

As an idea of how our findings may be used, think about the following Hilfer fractional neutral integro-differential inclusion,

$$D_{0^+}^{\frac{4}{7},\varepsilon} \left[ \Delta(\mathfrak{z}, v) - \overline{\mathcal{N}}(\mathfrak{z}, \Delta(\mathfrak{z}, v)) \right] \in \frac{\partial^2}{\partial \mathfrak{z}^2} \Delta(\mathfrak{z}, v) + \bar{\mathcal{G}}(\mathfrak{z}, \Delta(\mathfrak{z}, v), (E\Delta)(\mathfrak{z}, v)) \mathfrak{z} \in (0, d], v \in [0, \pi],$$

$$\Delta(\mathfrak{z}, 0) = \Delta(\mathfrak{z}, \pi) = 0 \mathfrak{z} \in [0, d], \tag{6}$$

$$I^{(1-\frac{4}{7})(1-\varepsilon)} y(w, 0) = y_0(v), v \in [0, \pi],$$

where $D_{0+}^{\frac{4}{7},\varepsilon}$ is the *HFD* of order $\frac{4}{7}$, type $\varepsilon$, $I^{(1-\frac{4}{7})(1-\varepsilon)}$ is the Riemann–Liouville integral of order $\frac{3}{7}(1-\varepsilon)$, $\bar{\mathcal{G}}\big(\mathfrak{z},\Delta(\mathfrak{z},v),(E\Delta)(\mathfrak{z},v)\big)$, $(E\Delta)(\mathfrak{z},v)$, and $\bar{\mathcal{N}}(\mathfrak{z},\Delta(\mathfrak{z},v))$ are the required functions.

To write the system (6) in the abstract form of (1)–(2), we chose the space $Y = L^2[0,\pi]$. Define an almost sectorial operator $\mathtt{A}$ by $\mathtt{A}\Delta = \Delta_{\mathfrak{z}\mathfrak{z}}$ with the domain

$$D(\mathtt{A}) = \left\{ \Delta \in Y : \frac{\partial \Delta}{\partial \mathfrak{z}}, \frac{\partial^2 \Delta}{\partial \mathfrak{z}^2} \in Y : \Delta(\mathfrak{z},0) = \Delta(\mathfrak{z},\pi) = 0 \right\}.$$

Then $\mathtt{A}$ produces a compact semigroup that is analytic and self-adjoint, $\mathtt{T}(\mathfrak{z})_{\mathfrak{z}} \geq 0$. Additionally, the discrete spectrum of $\mathtt{A}$ contains eigenvalues of $k^2, k \in \mathbb{N}$ and orthogonal eigenvectors $\zeta_k(z) = \sqrt{\frac{2}{\pi}}\sin(kz)$, then

$$\mathtt{A}z = \sum_{k=0}^{\infty} k^2 \langle z, \zeta_k \rangle \zeta_k.$$

Moreover, we have each $v \in Y$, $\mathtt{T}(\mathfrak{z})v = \sum_{k=1}^{\infty} \zeta^{-k^2\mathfrak{z}} \langle v, \zeta_k \rangle \zeta_k$. In particular, $\mathtt{T}(\cdot)$ is uniformly stable semigroup and $\|\mathtt{T}(\mathfrak{z})\| \leq M$, which satisfies $(H_1)$.

$\mathtt{y}(\mathfrak{z})(v) = \Delta(\mathfrak{z},v)$, $\mathfrak{z} \in \mathcal{J} = [0,d]$, $v \in [0,\pi]$. Take $\mathtt{y} \in Y = L^2[0,\pi]$, $v \in [0,\pi]$, we consider the multi-valued mapping $\mathcal{G} : \mathcal{J} \times Y \times Y \to Y$,

$$\mathcal{G}\big(\mathfrak{z},\mathtt{y}(\mathfrak{z}),(E\mathtt{y})(\mathfrak{z})\big) = \overline{\mathcal{G}}\big(\mathfrak{z},\Delta(\mathfrak{z},v),(E\Delta)(\mathfrak{z},v)\big)$$
$$= \frac{e^{-\mathfrak{z}}}{1+e^{-\mathfrak{z}}}\sin\left(w(\mathfrak{z},v) + \int_0^{\mathfrak{z}}\cos(\mathfrak{z}s)\Delta(s,v)ds\right),$$

where

$$(E\mathtt{y})(\mathfrak{z})(v) = \int_0^{\mathfrak{z}} e(\mathfrak{z},s,\Delta(s,v))ds = \int_0^{\mathfrak{z}}\cos(\mathfrak{z}s)\Delta(s,v)ds.$$

Since, mapping $\overline{\mathcal{G}}$ is measurable, upper semi-continuous, and strongly measurable,

$$\overline{\mathcal{G}}\big(\mathfrak{z},\Delta(\mathfrak{z},v),(E\Delta)(\mathfrak{z},v)\big) \leq \mathcal{M}_1^*.$$

So $\overline{\mathcal{G}}$ is satisfied $(H_2)$. Additionally, $\mathcal{N} : \mathcal{J} \times Y \to Y$ must have completely continuous mapping, which is defined as $\mathcal{N}(\mathfrak{z},u(\mathfrak{z})) = \overline{\mathcal{N}}(\mathfrak{z},\Delta(\mathfrak{z},v))$, satisfying the necessary hypotheses. Therefore, the required mapping satisfied all hypotheses. As a result, the nonlocal Cauchy problem (1)–(2) may be used to rephrase the fractional system (6). It is clear that the boundary of $\mathcal{G}\big(\mathfrak{z},\Delta(\mathfrak{z},u),(E\Delta)(\mathfrak{z},u)\big)$ is uniform. The problem has a mild solution on $\mathcal{J}$, according to Theorem 2 .

## 5. Conclusions

In this study, Martelli's fixed point theorem was used to examine the possibility of a mild solution for an abstract Hilfer fractional differential system via almost sectorial operators. Adequate criteria were applied to the present findings and were satisfied. The controllability of the Hilfer fractional neutral derivative (via almost sectorial operators) will be investigated in the future using a fixed point technique.

**Author Contributions:** Conceptualisation, C.B.S.V.B. and R.U.; methodology, C.B.S.V.B.; validation, C.B.S.V.B. and R.U.; formal analysis, C.B.S.V.B.; investigation, R.U.; resources, C.B.S.V.B.; writing original draft preparation, C.B.S.V.B.; writing review and editing, R.U.; visualisation, R.U.; supervision, R.U.; project administration, R.U. All authors have read and agreed to the published version of the manuscript.

**Funding:** There are no funders to report for this submission.

**Institutional Review Board Statement:** Not applicable.

**Informed Consent Statement:** Not applicable.

**Data Availability Statement:** Data sharing is not applicable to this article as no datasets were generated or analyzed during the current study.

**Acknowledgments:** The authors are grateful to the reviewers of this article who provided insightful comments and advice that allowed us to revise and improve the content of the paper. The first author would like to thank the management of VIT University for providing a teaching cum research assistant fellowship.

**Conflicts of Interest:** The authors have no conflict of interest to declare.

## Abbreviations

The following abbreviations are used in this manuscript:

| HFD | Hilfer fractional derivative |
| HF | Hilfer fractional |

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
