# Peer review of "Existence of Mild Solutions for Hilfer Fractional Neutral Integro-Differential Inclusions via Almost Sectorial Operators"

_fractalfract, doi:10.3390/fractalfract6090532_

Round 1
Reviewer 1 Report
1) The title of this paper should be modified. The authors should stress their innovations by comparing their results with recent results in literature.
2) In Theorem 3.1, the definition of $B_{P}(\mathcal{J})$ is not given.
3) Please check the formula in line 13 on page 7.
4) Please check step 2 in the proof of Theorem 3.1, pay attention to the estimate of
$\|S_{k,\varepsilon}(\mathfrak{z})[y_{0}-N(0, y(0)]\|$.
5) In the example, the authors should verify $(H_{1})$- $(H_{7})$ in details.
6) There are a lot of typos and grammatical mistakes in the manuscript, for example
i. We devote our interest in this manuscript to investigate the existence of Hilfer fractional neutral integro-differential systems with almost sectorial operator: "investigate" should be "investigating", "the existence of" should be "the existence of mild solution".
ii. Martelli's fixed point theorem are used to prove the results: "are used" should be "is used".
iii. In line 15 on page 7, "We have N has convex values" .
\item In line 10 on page 7, a comma and a "for" are missing.
\item In line 18 on page 7, "$\psi_{2}$" should be "$\psi$".
\item In the last line of page 14, "if" should be "of".

Author Response
The authors are very much grateful to the respected reviewer for their efforts and detailed comments on the paper which very much helped considerably to revise and improve the paper. We have incorporated all the comments making appropriate changes as suggested by the reviewer. The authors thank the reviewer for the time spent providing a very helpful review with meticulous comments. With effort, we have revised the manuscript according to the advice from the reviewer. Please find the attachment.

Reviewer 2 Report
The paper deals with the existence of Hilfer fractional neutral integro-differential systems with almost sectorial operator. Martelli’s fixed point theorem is used to prove the primary results. The paper has good merit. However I have some concerns including the writing style of the paper.
1. The english has to be fixed throughout the paper. Sentences like "Martelli’s fixed point theorem are used"(?) should be made grammatically correct. Also, equations should be appropriately punctuated (example: last line at the end of Step 2 on page 8, and many more!).
2. I am not sure why quotation marks are used in the statements of definitions/lemmas/theorems in Section 2. This is very unusual, unless it is a requirement by the journal.
3. Section 3: what are the basis of all the hypotheses? Are those optimal? Can some of those be reduced?
4. Most of the computations in Step 3 (page 8) are redundant. Those should be omitted.
5. I am confused about Step 4. Can you please clarify? For example- the first line on page 12. Also, what does the first sentence of Step 5 mean? What is the context? are you trying to prove that the graph is closed?
6. Is there any comparable method in the literature? In other words, why is the example in Section 4 significant?
7. Fractional integro-differential system has many applications in finance. Surprisingly that is missing in the introduction. For example, the authors are requested to check the paper and cite if appropriate: "Fractional Barndorff-Nielsen and Shephard model: applications in variance and volatility swaps, and hedging, Annals of Finance, Vol. 17, 2021, pp. 529–558" and references therein.
Once these are addressed I will be happy to read the revised version.
Author Response

(The authors gave the same response as above.)

Round 2
Reviewer 1 Report
I recommend this manuscript for publication in it's current revised form.
Reviewer 2 Report
The paper addresses all my concerns. It is improved.